# Stochastic Proximal Point Algorithm for Large-scale Nonconvex Optimization: Convergence, Implementation, and Application to Neural Networks

## Abstract

We revisit the stochastic proximal point algorithm (SPPA) for large-scale nonconvex optimization problems. SPPA has been shown to converge faster and more stable than the celebrated stochastic gradient descent (SGD) algorithm, and its many variations, for convex problems. However, the per-iteration update of SPPA is defined abstractly and has long been considered expensive. In this paper, we show that efficient implementation of SPPA can be achieved. If the problem is a nonlinear least squares, each iteration of SPPA can be efficiently implemented by Gauss-Newton; with some linear algebra trick the resulting complexity is in the same order of SGD. For more generic problems, SPPA can still be implemented with L-BFGS or accelerated gradient with high efficiency. Another contribution of this work is the convergence of SPPA to a stationary point in expectation for nonconvex problems. The result is encouraging that it admits more flexible choices of the step sizes under similar assumptions. The proposed algorithm is elaborated for both regression and classification problems using different neural network structures. Real data experiments showcase its effectiveness in terms of convergence and accuracy compared to SGD and its variants.

## 1 Introduction

Algorithm design for large-scale machine learning problems have been dominated by the stochastic (sub)gradient descent (SGD) and its variants (Bottou et al., 2018). The main reasons are two-fold: on the one hand, the size of the data set may be so large that obtaining the full gradient information is too costly; on the other hand, solving the formulated problem to very high accuracy is typically unnecessary in machine learning, since the ultimate goal of most tasks is not to fit the training data but to generalize well on unseen data. As a result, stochastic algorithms such as SGD has gained tremendous popularity recently.

There has been many variations and extensions of the plain vanilla SGD algorithm to accelerate its convergence rate. One line of research focuses on reducing the variance of the stochastic gradient, resulting in famous algorithms such as SVRG (Johnson and Zhang, 2013) and SAGA (Defazio et al., 2014), which results in extra time/memory complexities of the algorithm (significantly). More recently, adaptive learning schemes such as AdaGrad (Duchi et al., 2011) and Adam (Diederik P. Kingma, 2014) have shown to be more effective in keeping the algorithm fully stochastic and light-weight. In terms of theory, there has also been surging amount of work quantifying the best possible rate using first-order information (Lei et al., 2017; Allen-Zhu, 2017; 2018a;b), as well as its ability to obtain not only stationary points but also local optima (Ge et al., 2015; Jin et al., 2017; Xu et al., 2018; Allen-Zhu, 2018).

### 1.1 Stochastic proximal point algorithm (SPPA)

In this work, we consider a different type of stochastic algorithm called the stochastic proximal point algorithm (SPPA), also known as incremental proximal point method (Bertsekas, 2011a;b) or stochastic proximal iterations (Ryu and Boyd, 2014). Consider the following optimization problem

with the objective function in the form of a finite sum of component functions

$$\underset{\boldsymbol{\theta} \in \mathbb{R}^d}{\text{minimize}} \quad \frac{1}{n} \sum_{i=1}^{n} \ell_i(\boldsymbol{\theta}) = L(\boldsymbol{\theta}). \tag{1}$$

SPPA takes the following simple form:

1: **repeat**
2:     randomly draw $i$ from $\{1, \ldots, n\}$
3:     $\boldsymbol{\theta}_{t+1} \leftarrow \arg\min_{\boldsymbol{\theta}} \lambda_t \ell_i(\boldsymbol{\theta}) + (1/2)\|\boldsymbol{\theta} - \boldsymbol{\theta}_t\|^2 = \text{Prox}_{\lambda_t \ell_i}(\boldsymbol{\theta}_t)$
4: **until** convergence

The update rule in line 3 is called the proximal operator of the function $\lambda_t \ell_i$ evaluated at $\boldsymbol{\theta}_t$. This is the stochastic version of the proximal point algorithm, which dates back to Rockafellar (1976).

Admittedly, SPPA is not as universally applicable as SGD, due to the abstraction of the per-iteration update rule. It is also asking for more information from the problem than merely the first-order derivatives. However, with the help of more information inquired, there is also hope that it provides faster and more robust convergence guarantees.

> As we will see in numerical experiments, SPPA is able to achieve good optimization performance by taking fewer number of passes through the data set, although it takes a little more computations for each batch. We believe in many cases it is worth trading off more computations for fewer memory accesses.

To the best of our knowledge, convergence analyses of SPPA has only been studied for convex problems (Bertsekas, 2011a; Ryu and Boyd, 2014; Bianchi, 2016). Their study shows that SPPA converges somewhat similar to SGD for convex problems, but the updates are much more robust to instabilities in the problem. Most authors also accept the premise that the proximal operator is sometimes difficult to evaluate, and thus proposed variations to the plain vanilla version to handle more complicated problem structures (Wang and Bertsekas, 2013; Duchi and Ruan, 2018; Asi and Duchi, 2019b; Davis and Drusvyatskiy, 2019).

In terms of nonconvex optimization problems, there is very little work until very recently (Davis and Drusvyatskiy, 2019; Asi and Duchi, 2019a). However, their convergence analysis is somewhat unconventional. Typically for a nonconvex problem, we would expect a theoretical claim that the iterates generated by SPPA converges (in expectation) to a stationary point. This is not easy, and the result given by (Davis and Drusvyatskiy, 2019) and (Asi and Duchi, 2019a) defined an imaginary sequence (that is not computed in practice) $\{\widetilde{\boldsymbol{\theta}}_t\}$ as $\widetilde{\boldsymbol{\theta}}_t = \arg\min_{\boldsymbol{\theta}} \lambda_t L(\boldsymbol{\theta}) + (1/2)\|\boldsymbol{\theta} - \boldsymbol{\theta}_t\|^2$, i.e., the proximal operator *of the full loss function* from the algorithm sequence $\{\boldsymbol{\theta}_t\}$. Their results show that this imaginary sequence $\{\widetilde{\boldsymbol{\theta}}_t\}$ converges to a stationary point in expectation.

## 1.2 CONTRIBUTIONS

There are two main contributions we present in this paper; efficient implementations of proximal operator update for SPPA with application on regression and classification problems, and convergence to a stationary point for SPPA algorithm for general non-convex problems. The cost of this abstract per-iteration update rule has been the burden for SPPA algorithm, even though it has been shown to converge faster and more stable than the celebrated SGD algorithm. In this paper, we show that it is actually not a burden when the per-iteration update is efficiently implemented.

In the implementation section we will discuss the implementation of abstract per-iteration update rule for non-linear least squares (NLS) problem and other non-linear problems. We present two different implementations for these two different categories of problems. SPPA-Gauss Newton (SPPA-GN) and SPPA-Accelerated, respectively for regression with nonlinear least squares and classification problems. We apply SPPA to a large family of nonconvex optimization problems and show that the seemingly complicated proximal operator update can still be efficiently obtained. Both implementations give results that are comparable with the state of the art stochastic algorithms.

On the other hand, there is large group of nonlinear problems that are not necessarily expressed as a NLS problem (non-NLS). Many of the classification problems in deep learning applications as of today, use different type of loss functions than mean least squared error. Hence, we suggest an

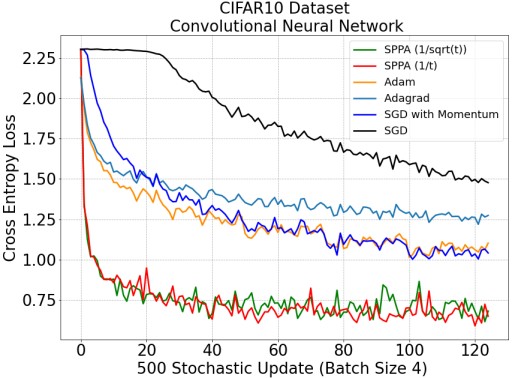

Figure 1: CIFAR10: cross entropy loss

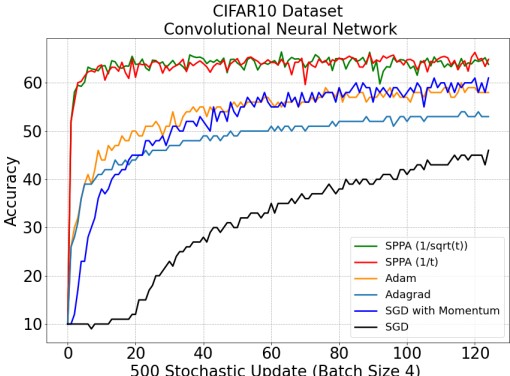

Figure 2: CIFAR10: prediction accuracy

alternative to SPPA-GN where we implement the per-iteration update rule based on well-known stochastic optimization algorithms, preferably with the ones that take less number of iterations to converge like L-BFGS (Liu and Nocedal, 1989).

As a second contribution, we show that SPPA, when applied to a nonconvex problem, converges to a stationary point in expectation, as informally stated as follows

**Theorem 1.** *(informal.) For SPPA with fixed* $\lambda_t = \lambda$*, the expected gradient converges to a region where the norm is bounded with radius proportional to* $\lambda$*. For diminishing step sizes such that*

$$\lim_{T \to \infty} \frac{T}{\sum_{t=1}^{T} \lambda_t^{-1}},$$

*the expected gradient converges to zero.*

Finally, we apply SPPA with the novel efficient updates to some classical regression and classification using 4 different data sets for neural network training. Detailed algorithmic descriptions are provided, and we show the outstanding performance of SPPA when effectively executed.

As a sneak peek of the numerical performance of the proposed SPPA implementation, Figures 1 and 2 shows the cross entropy loss and prediction error on the test set over the progression of SPPA verses various baseline algorithms. SPPA indeed converges much faster than all SGD-based methods, achieving peak prediction accuracy after only going through the data set approximately 10 times. We should stress that the per-iteration complexity of SPPA is higher than other methods, since it tries to solve a small optimization problem rather than a simple gradient update. However, as we have argued before, in many cases it is worth trading off more computations for fewer memory accesses.

## 2 EFFICIENT IMPLEMENTATION

In this section we introduce several efficient methods to calculate the proximal operator update given in pseudo-code of SPPA section 1.1 line 3. The two methods emerge from the question of 'How can we implement the proximal operator update efficiently?'. The answer to this question depends on the type of the objective function. We considered the problems in two different categories, nonlinear least squares (NLS) and other generic nonlinear problems.

### 2.1 SPPA-GN FOR NONLINEAR LEAST SQUARES

A nonlinear least squares (NLS) problem takes the following form

$$\underset{\boldsymbol{\theta} \in \mathbb{R}^d}{\text{minimize}} \quad \frac{1}{n} \sum_{i=1}^{n} \frac{1}{2} (\varphi_i(\boldsymbol{\theta}))^2, \tag{2}$$

where each $\varphi_i$ is a general nonlinear function with respect to $\boldsymbol{\theta}$. It is a classical nonlinear programming problem (Bertsekas, 1999; Boyd and Vandenberghe, 2018) with many useful applications, including

least squares neural networks (Van Der Smagt, 1994), where each $\varphi_i$ corresponds to the residual of fitting for the $i$th data sample for regression. We will show that problems of this form can be efficiently executed by SPPA-GN, despite its seemingly complication.

To apply SPPA to NLS, the main challenge is to efficiently evaluate the proximal operator

$$\boldsymbol{\theta}_{t+1} \leftarrow \arg\min_{\boldsymbol{\theta}} \frac{\lambda_t}{2}(\varphi_i(\boldsymbol{\theta}))^2 + \frac{1}{2}\|\boldsymbol{\theta} - \boldsymbol{\theta}_t\|^2. \tag{3}$$

Notice that this function itself is a nonlinear least squares objective, although with only one component function together with the proximal term.

The traditional wisdom to solve a NLS problem is to apply the Gauss-Newton (GN) algorithm: at each iteration, we first take a first-order approximation of the vector-valued function inside the Euclidean norm, and set the update as the solution of the approximated linear least squares problem. It is a well-known algorithm that can be found in many standard textbooks, e.g., (Bertsekas, 1999; Nocedal and Wright, 2006; Boyd and Vandenberghe, 2018).

To apply GN to (3), we first take linear approximation of $\varphi_i$ at the current update $\overline{\boldsymbol{\theta}}$ as

$$\varphi_i(\boldsymbol{\theta}) \approx \varphi_i(\overline{\boldsymbol{\theta}}) + \nabla\varphi_i(\overline{\boldsymbol{\theta}})^\top(\boldsymbol{\theta} - \overline{\boldsymbol{\theta}}),$$

and set the solution of the following problem $\boldsymbol{\theta}^+$ as the next update

$$\underset{\boldsymbol{\theta}}{\text{minimize}} \quad \frac{\lambda_t}{2}(\varphi_i(\overline{\boldsymbol{\theta}}) + \nabla\varphi_i(\overline{\boldsymbol{\theta}})^\top(\boldsymbol{\theta} - \overline{\boldsymbol{\theta}}))^2 + \frac{1}{2}\|\boldsymbol{\theta} - \boldsymbol{\theta}_t\|^2. \tag{4}$$

Obviously, (4) has a closed form solution

$$\boldsymbol{\theta}^+ = \boldsymbol{\theta}_t - \left(\frac{1}{\lambda_t}\boldsymbol{I} + \boldsymbol{g}_i\boldsymbol{g}_i^\top\right)^{-1}\boldsymbol{g}_i(\varphi_i(\overline{\boldsymbol{\theta}}) - \boldsymbol{g}_i^\top(\overline{\boldsymbol{\theta}} - \boldsymbol{\theta}_t)), \tag{5}$$

where we denote $\boldsymbol{g}_i = \nabla\varphi_i(\overline{\boldsymbol{\theta}})$ to simplify notation. Notice that the matrix to be inverted in (5) has a simple "identity plus rank-one" structure, implying that it can be efficiently computed in linear time. Using the "kernel trick" (Boyd and Vandenberghe, 2018, pp.332)

$$(\boldsymbol{A}^\top\boldsymbol{A} + \alpha\boldsymbol{I})^{-1}\boldsymbol{A}^\top = \boldsymbol{A}^\top(\boldsymbol{A}\boldsymbol{A}^\top + \alpha\boldsymbol{I})^{-1}, \tag{6}$$

update (5) simplifies to

$$\boldsymbol{\theta}^+ = \boldsymbol{\theta}_t - \frac{\varphi_i(\overline{\boldsymbol{\theta}}) - \nabla\varphi_i(\overline{\boldsymbol{\theta}})^\top(\overline{\boldsymbol{\theta}} - \boldsymbol{\theta}_t)}{\lambda_t^{-1} + \|\nabla\varphi_i(\overline{\boldsymbol{\theta}})\|^2}\nabla\varphi_i(\overline{\boldsymbol{\theta}}). \tag{7}$$

As we can see, each GN update only takes $\mathcal{O}(d)$ flops, which is as cheap as that of a SGD step. To fully obtain the proximal operator (3), one has to run GN for several iterations. However, thanks to the superlinear convergence rate of GN near its optimal (Nocedal and Wright, 2006), which is indeed the case if we initiate at $\boldsymbol{\theta}_t$ because of the proximal term, it typically takes no more than 5–10 GN updates.

The detailed description of the proposed algorithm, which we term SPPA-GN, for solving general NLS problems is shown in Algorithm 1.

---

**Algorithm 1** SPPA-GN

1: initialize $\boldsymbol{\theta}_0, t \leftarrow 0$
2: **repeat**
3:     randomly draw $i$ from $\{1, \ldots, n\}$
4:     $\boldsymbol{\theta}^+ \leftarrow \boldsymbol{\theta}_t$
5:     **repeat**
6:         $\overline{\boldsymbol{\theta}} \leftarrow \boldsymbol{\theta}^+$
7:         $\boldsymbol{\theta}^+ \leftarrow \boldsymbol{\theta}_t - \frac{\varphi_i(\overline{\boldsymbol{\theta}}) - \nabla\varphi_i(\overline{\boldsymbol{\theta}})^\top(\overline{\boldsymbol{\theta}} - \boldsymbol{\theta}_t)}{\lambda_t^{-1} + \|\nabla\varphi_i(\overline{\boldsymbol{\theta}})\|^2}\nabla\varphi_i(\overline{\boldsymbol{\theta}})$
8:     **until** convergence
9:     $\boldsymbol{\theta}_{t+1} \leftarrow \boldsymbol{\theta}^+, t \leftarrow t + 1$
10: **until** convergence

---

## 2.2 SPPA-LBFGS AND SPPA-AGD FOR GENERIC LOSSES

Even for generic nonlinear programming problems, it is still possible to efficiently evaluate the proximal update beyond merely a simple gradient step. The idea is to apply the limited-memory BFGS algorithm (Nocedal and Wright, 2006), or L-BFGS for short. L-BFGS is a memory-efficient

implementation of the famous quasi-Newton algorithm BFGS. In a nut shell, L-BFGS is an iterative algorithm that makes use of the second-order information from the optimization loss function, but does not require solving matrix inverses and only requires explicitly evaluating first-order gradients. For a prescribed number of iterations, it requires one matrix-vector multiplication and multiple vector multiplications. As a result, the overall complexity is again $\mathcal{O}(d)$ if the initial guess of the Hessian matrix is diagonal.

While there are certain limitations for applying L-BFGS to general nonlinear programming problems, we reckon that it fits perfectly in the context of SPPA implementations.

- L-BFGS has to specify a good initial guess of the Hessian approximation matrix. While in many cases people simply use the identity matrix to start, it may result in very poor approximation. Fortunately, thanks to the proximal term, the identity matrix is in fact a very good initial guess for the Hessian matrix for SPPA updates.
- In order to save memory consumption, L-BFGS has to prescribe the number of iterations before running the algorithm. Obviously, if the prescribed number of iteration is too large, we incur unnecessary computations, while if it is too small we need to invoke another round with a new estimated Hessian matrix. However, again in the context of SPPA, the proximal term $\|\boldsymbol{\theta} - \boldsymbol{\theta}_t\|^2$ naturally provides a good initialization $\boldsymbol{\theta}_t$. Our experience show that prescribing 10 iterations of L-BFGS updates is more than enough to obtain extremely accurate solutions.

On the other hand, it perhaps makes more sense to simply use some more advanced first-order methods such as Nesterov's accelerated gradient descent (1983; 2013) to calculate the proximal update. Both L-BFGS and accelerated gradient descent evaluates the gradient of the loss function with $\mathcal{O}(d)$ complexity, and the question is how to leverage convergence rate versus sophistication.

## 3 CONVERGENCE ANALYSIS

In this section we provide convergence analysis of SPPA for general nonconvex loss functions (1) to a stationary point, as informally stated in Theorem 1. To the best of our knowledge, convergence of SPPA for nonconvex problems is still missing as of this writing.

There is a well-known resemblance between proximal methods and gradient descent: while a full gradient descent step takes the form $\boldsymbol{\theta}_{t+1} = \boldsymbol{\theta}_t - \lambda_t \nabla L(\boldsymbol{\theta}_t)$, the definition of a full proximal step guarantees that $\lambda_t \nabla L(\boldsymbol{\theta}_{t+1}) = \boldsymbol{\theta}_{t+1} - \boldsymbol{\theta}_t$, meaning that $\boldsymbol{\theta}_{t+1} = \boldsymbol{\theta}_t - \lambda_t \nabla L(\boldsymbol{\theta}_{t+1})$. Therefore, one might expect that a well-established convergence analysis of SGD for nonconvex problems, for example (Bottou et al., 2018, §4.3), can be seamlessly applied to SPPA. This is unfortunately not the case.

Consider $\mathrm{E}[\|\nabla L(\boldsymbol{\theta}_t)\|^2]$, where the expectation is taken over the sampling procedure conditioned on $\boldsymbol{\theta}_t$, for SGD we typically require $\nabla \ell_i(\boldsymbol{\theta}_t)$ to be an unbiased estimator of $\nabla L(\boldsymbol{\theta}_t)$, which is easy to satisfy. For SPPA then one needs to consider $\mathrm{E}[\|\nabla L(\boldsymbol{\theta}_{t+1})\|^2]$, again over the sampling procedure conditioned on $\boldsymbol{\theta}_t$. This is in fact difficult to quantify because the update $\boldsymbol{\theta}_{t+1}$ depends on the sample that is drawn from the data set. It requires a somewhat different approach to establish its convergence.

The convergence analysis is developed under the following three assumptions, and we explain their implications when applied to SPPA. The detailed proofs are presented in supplementary document.

**Assumption 1.** The loss function (1) has bounded deviation: $|\mathrm{E}[L(\boldsymbol{\theta}) - \ell_i(\boldsymbol{\theta})]| \leq \sigma$.

**Assumption 2.** Each component function $\ell_i(\boldsymbol{\theta})$ is differentiable and Lipschitz continuous with parameter $\mu$: $\|\nabla \ell_i(\boldsymbol{\theta}) - \nabla \ell_i(\tilde{\boldsymbol{\theta}})\| \leq \mu \|\boldsymbol{\theta} - \tilde{\boldsymbol{\theta}}\|, \qquad \forall \boldsymbol{\theta}, \tilde{\boldsymbol{\theta}} \in \mathrm{dom}\, \ell_i$.

**Assumption 3.** The SPPA updates lie in a compact set, i.e., there exists a constant $c$ that $\|\boldsymbol{\theta} - \tilde{\boldsymbol{\theta}}\| \leq c$ for all $\boldsymbol{\theta}$ and $\tilde{\boldsymbol{\theta}}$.

Under these rather standard assumptions, we have the following proposition, which serves as the stepping stone for our main convergence results. The proof of Proposition 1 is relegated to the appendix.

**Proposition 1.** *Under Assumptions 1, 2 and 3, at iteration $t$ we have*

$$\|\nabla L(\boldsymbol{\theta}_t)\|^2 \leq 4\mu^2 \lambda_t \left( L(\boldsymbol{\theta}_t) - \mathrm{E}[L(\boldsymbol{\theta}_{t+1})] + \sigma \right) + \lambda_t C, \tag{8}$$

*where $C = c^2(n+1)$ and the expectation is taken over the sampling of $i$ conditioned on $\boldsymbol{\theta}_t$.*

With the help of Proposition 1, the rest of the results follows similarly to those presented in (Bottou et al., 2018, §4.3) for constant step sizes and diminishing step sizes. However, we point out two significant advantages compared to the standard SGD results:

- For constant step sizes, there is no limit on how small $\lambda$ should be. Obviously a larger $\lambda$ rapidly converges to a bigger vicinity around a stationary point, while a smaller $\lambda$ slowly converges to a smaller vicinity. However, unlike SGD, the constant step size $\lambda$ can be arbitrarily large, and the expected gradient can still be bounded.

- For diminishing step sizes, we only need $\lambda_t \to 0$ and $\sum \lambda_t \to \infty$, but not the square summable assumption $\sum \lambda_t^2 < \infty$. This gives us the option of picking a slower diminishing step size such as $1/\sqrt{t}$ for faster convergence rate.

### 3.1 CONVERGENCE WITH CONSTANT STEP SIZES

**Theorem 2.** *Under Assumptions 1, 2 and 3, suppose SPPA is run with a constant step size $\lambda_t = \lambda$ for all t, then the expected averaged squared gradients satisfy*

$$\mathrm{E}\left[\frac{1}{T}\sum_{t=1}^{T}\|\nabla L(\boldsymbol{\theta}_t)\|^2\right] \leq 4\mu^2\lambda\left(\frac{L(\boldsymbol{\theta}_0) - L_{\inf}}{T} + \sigma\right) + \lambda C \xrightarrow{T\to\infty} \lambda(4\mu^2\sigma + C) \tag{9}$$

*Proof.* Taking the total expectation of (8) and summing over $t = 1, \ldots, T$, we get

$$\mathrm{E}\left[\sum_{t=1}^{T}\|\nabla L(\boldsymbol{\theta}_t)\|^2\right] \leq 4\mu^2\lambda\left(L(\boldsymbol{\theta}_0) - \mathrm{E}[L(\boldsymbol{\theta}_T)] + T\sigma\right) + T\lambda C.$$

Replace $\mathrm{E}[L(\boldsymbol{\theta}_T)]$ with the infimum $L_{\inf} \leq \mathrm{E}[L(\boldsymbol{\theta}_T)]$ and divide both sides by $T$ shows (9). □

Theorem 2 shows that as $T$ increases, SPPA spends increasingly more time in regions where the updates have relatively small gradients. The size of the region is proportional to $\lambda$. The interesting observation is that $\lambda$ can be arbitrarily large, unlike the SGD case, albeit the bound on the expected gradient would also be thereupon large as well.

### 3.2 CONVERGENCE WITH DIMINISHING STEP SIZES

To further drive the expected gradients closer to zero, we can use diminishing step sizes as shown below. Notice that we only require that the sum of the step size *reciprocals* to grow faster than $T$, without the more stringent condition that the diminishing step sizes are square summable. This allows us to use step sizes that decrease a lot slower, such as $\lambda_t = 1/\sqrt{t}$, for faster convergence rate, while ensuring that the expected gradient goes to zero asymptotically. An analogous result has not been achieved for SGD when applied to nonconvex problems.

**Theorem 3.** *Under Assumptions 1, 2 and 3, suppose SPPA is run with a diminishing step size, then the expected averaged squared gradients satisfy*

$$\mathrm{E}\left[\frac{1}{A_T}\sum_{t=1}^{T}\alpha\|\nabla L(\boldsymbol{\theta}_t)\|^2\right] \leq \frac{4\mu^2\left(L(\boldsymbol{\theta}_0) - L_{\inf} + T\sigma\right) + TC}{A_T} \xrightarrow{T\to\infty} 0, \tag{10}$$

*where $\alpha_t = 1/\lambda_t$ and $A_T = \sum_{t+1}^{T}\alpha_t$, if the step sizes satisfy that*

$$\lim_{T\to\infty}\frac{T}{A_T} = 0. \tag{11}$$

*Proof.* We rearrange (8) into

$$\lambda_t^{-1}\|\nabla L(\boldsymbol{\theta}_t)\|^2 \leq 4\mu^2\left(L(\boldsymbol{\theta}_t) - \mathrm{E}[L(\boldsymbol{\theta}_{t+1})] + \sigma\right) + C.$$

Then apply similar strategy as in the proof of Theorem 2 to obtain the first line of (10). Furthermore, if (11) holds, (10) goes to zero as $T \to \infty$. □

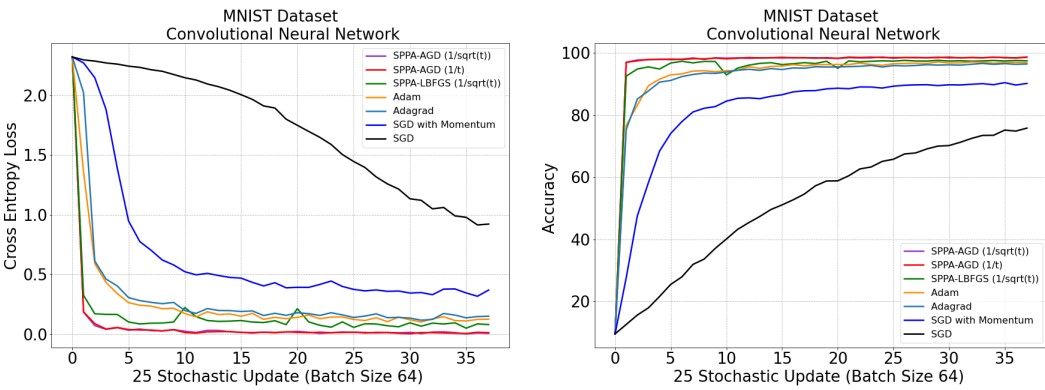

Figure 3: MNIST: cross entropy loss        Figure 4: MNIST: prediction accuracy

## 4 EXPERIMENTS

We show some real data experiments to demonstrate effectiveness of the proposed SPPA implementations in classification and regression problems respectively. We compare our proposed algorithms with three variants of SGD: the original version with various step strategies, SGD with momentum (Sutskever et al., 2013), Adagrad (Duchi et al., 2011), and Adam (Diederik P. Kingma, 2014), with the default settings suggested in their original papers. We used the PyTorch (Paszke et al., 2019) implementation of the algorithms and used the same platform for implementing our customized optimization algorithm.

### 4.1 CLASSIFICATION WITH SPPA-AGD AND SPPA-LBFGS

We present the results of the classification problem with two data sets (MNIST (LeCun et al., 2010), CIFAR10 (Krizhevsky et al., 2010)) using SPPA-AGD algorithm. In all the results presented, the settings are the default settings provided by PyTorch library. We experimented with different settings with all the algorithms and the best results are obtained when the settings were the default settings.

**MNIST** MNIST Handwritten Digits Dataset (LeCun et al., 2010) is a common dataset used for showing the effectiveness of many state of art the algorithms. It consists of 60,000 training, 10,000 test images of size $28 \times 28$ in 10 classes. The network we used for this experiment is based on convolutional deep neural network. It consists of 4 layers, the first 2 of which being convolutional and last 2 being fully connected. The loss function is cross entropy loss, and the activation function is RELU. Batch size is 64. To show the behaviour of the algorithms more in detail we recorded the loss values at every 25 stochastic update, the experiment is run for 1 epoch and in total 38 updates are presented ($25 \times 38 \times 64$ giving the number of training samples). The accuracy is calculated on the test data at every 500 stochastic update, using the accuracy calculation in PyTorch tutorial[1]. As observed in Figures 3 and 4, SPPA-based methods outperforms all SGD-based algorithms in terms of both the cross entropy loss and prediction accuracy on the test set.

**CIFAR10** CIFAR10 (Krizhevsky et al., 2010) is relatively extensive dataset compared to MNIST. It consists of 50,000 training, 10,000 test images of size $32 \times 32$ in 10 classes. The network we used for this experiment is based on convolutional deep neural network. It consists of 5 layers, the first 2 of which being convolutional and last 3 being fully connected. The loss function is cross entropy loss, and the activation function is RELU. Batch size is 4. To show the behaviour of the algorithms more in detail we recorded the loss values at every 500 stochastic update, the experiment is run for 5 epochs and in total 125 updates are presented ($25 \times 500 \times 4$ giving the number of training samples). The accuracy is calculated on the test data at every 500 stochastic update overall for 5 epochs, using the accuracy calculation in PyTorch tutorial. Convergence plots are shown in Figures 1 and 2 at the beginning of the paper.

---

[1]`https://github.com/pytorch/tutorials/blob/master/beginner_source/`
`blitz/cifar10_tutorial.py`

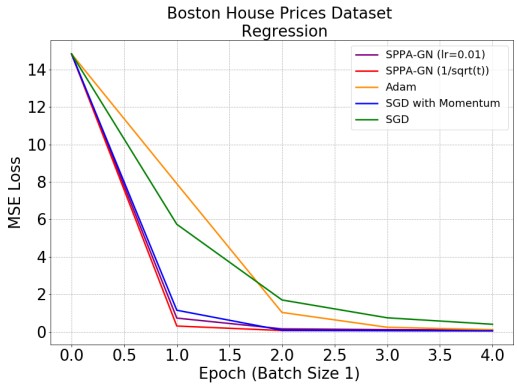

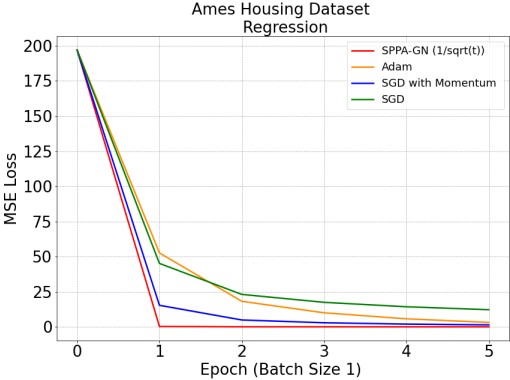

Figure 5: Boston housing prices          Figure 6: Ames housing prices

## 4.2 REGRESSION WITH SPPA-GN

In this subsection, we compare the performance of SPPA in two regression problems. Using the least squares loss, the problem becomes an instance of nonlinear least squares, thus our proposed SPPA-GN algorithm could be applied. The inner loop in SPPA-GN algorithm is run at most 10 times for the results presented below.

**Boston House Prices** Boston House Prices data set (Harrison Jr and Rubinfeld, 1978) is a relatively small data set but one of the most commonly used one for regression. It consists of 506 samples (404 training, 102 test ) with 13 features each to predict housing prices. The neural network used for this experiment is a multi-layer feed-forward network. It consists of 1 input layer, 2 hidden layers, and 1 output layer (13-(10-10)-1). All layers are fully connected. The batch size is 1. We present the loss value that we observed in every epoch. Figure 5 shows the test error obtained after each pass of the entire training set. Similar to the previous experiments, SPPA converges faster than all of the baseline SGD-based algorithms, although the improvement is not as great as the previous experiments due to the small-size of this data set.

**Ames Housing Prices** Ames Housing Dataset (De Cock, 2011) is an extended version of Boston House Prices data set (Harrison Jr and Rubinfeld, 1978). It consists of 1460 samples (1168 training, 292 test) with 80 features. The neural network used for this experiment is again a multi-layer feed-forward network. It consists of 1 input layer, 2 hidden layers, and 1 output layer (288-(72-18)-1). All layers are fully connected. The batch size is 1. Figure 6 shows the test error obtained after each pass of the entire training set. Again SPPA performs better than all other methods on this slightly bigger data set for regression. A somewhat remarkable observation is that the testing loss manages to achieve the minimum in only one pass of the entire data set (of course with a little more computations when handling each data point). If the computing node prefers more computations than data loading, SPPA is much more preferable in this case.

## 5 CONCLUSION

In this paper we presented efficient implementations of the stochastic proximal point algorithm (SPPA) for large-scale nonconvex learning problems. The specific approach to efficiently evaluate the proximal operators is based on the somewhat forgotten quasi-Newton methods with superlinear convergence rate such as Gauss-Newton and L-BFGS. We also proved that SPPA converges, in expectation, to a stationary point of a nonconvex problem with more flexible choices of the step sizes. Detailed proof can be found in the appendix. The resulting algorithm has cheap per-iteration complexity, while enjoying faster and more robust convergence under milder conditions. We demonstrate the performance of SPPA with our proposed efficient implementation on some well-known classification and regression data sets for neural network training, and showed that they outperform many existing methods.

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

## A    APPENDIX: PROOF OF PROPOSITION 1

First, we introduce a new notation

$$\boldsymbol{\theta}_{t+1}^{(i)} = \arg\min_{\boldsymbol{\theta}} \lambda_t \ell_i(\boldsymbol{\theta}) + \frac{1}{2}\|\boldsymbol{\theta} - \boldsymbol{\theta}_t\|^2 \tag{12}$$

to denote the update at iteration $t$ had we sampled the $i$th data point. Since $\ell_i$ is nonconvex, there is typically no guarantee that the exact minimum (12) is obtained. However, as we will see soon, the convergence analysis stays valid as long as the following two conditions hold:

- decrease:
$$\frac{1}{2\lambda_t}\|\boldsymbol{\theta}_{t+1}^{(i)} - \boldsymbol{\theta}_t\|^2 \le \ell_i(\boldsymbol{\theta}_t) - \ell_i(\boldsymbol{\theta}_{t+1}^{(i)}); \tag{13}$$

- stationarity:
$$\lambda_t \nabla \ell_i(\boldsymbol{\theta}_{t+1}^{(i)}) = \boldsymbol{\theta}_t - \boldsymbol{\theta}_{t+1}^{(i)}. \tag{14}$$

**Lemma 1.** *Under Assumption 1, we have at iteration $t$*

$$\mathrm{E}[\|\boldsymbol{\theta}_{t+1} - \boldsymbol{\theta}_t\|^2] \le 2\lambda_t \left(L(\boldsymbol{\theta}_t) - \mathrm{E}[L(\boldsymbol{\theta}_{t+1})] + \sigma\right), \tag{15}$$

*where expectation is taken over the sampling of $i$ conditioned on $\boldsymbol{\theta}_t$.*

*Proof.* Taking expectation of (13) over $i$ conditioned on $\boldsymbol{\theta}_t$ gets

$$\mathrm{E}[\|\boldsymbol{\theta}_{t+1} - \boldsymbol{\theta}_t\|^2] = \frac{1}{n}\sum_{i=1}^n \|\boldsymbol{\theta}_{t+1}^{(i)} - \boldsymbol{\theta}_t\|^2 \le 2\lambda_t \left(L(\boldsymbol{\theta}_t) - \frac{1}{n}\sum_{i=1}^n \ell_i(\boldsymbol{\theta}_{t+1}^{(i)})\right).$$

Invoking Assumption 1 shows (15). □

**Lemma 2.** *Under Assumption 2 and 3, we have at iteration $t$*

$$\|\nabla L(\boldsymbol{\theta}_t)\|^2 \le 2\mu^2 \mathrm{E}[\|\boldsymbol{\theta}_{t+1} - \boldsymbol{\theta}_t\|^2] + 2C^2, \tag{16}$$

*where expectation is taken over the sampling of $i$ conditioned on $\boldsymbol{\theta}_t$.*

*Proof.* Apply Assumption 2 to $\boldsymbol{\theta}_{t+1}^{(i)}$ and $\boldsymbol{\theta}_t$, and take expectation over $i$ conditioned on $\boldsymbol{\theta}_t$, we get

$$\frac{1}{n}\sum_{i=1}^n \|\nabla \ell_i(\boldsymbol{\theta}_{t+1}^{(i)}) - \nabla \ell_i(\boldsymbol{\theta}_t)\| \le \frac{\mu}{n}\sum_{i=1}^n \|\boldsymbol{\theta}_{t+1}^{(i)} - \boldsymbol{\theta}_t\|. \tag{17}$$

The left-hand-side of (17) can be lowerbounded by the triangle inequality

$$\left\|\frac{1}{n}\sum_{i=1}^n \nabla \ell_i(\boldsymbol{\theta}_{t+1}^{(i)}) - \nabla L(\boldsymbol{\theta}_t)\right\| \le \frac{1}{n}\sum_{i=1}^n \|\nabla \ell_i(\boldsymbol{\theta}_{t+1}^{(i)}) - \nabla \ell_i(\boldsymbol{\theta}_t)\| \tag{18}$$

where

$$\nabla L(\boldsymbol{\theta}_t) = \frac{1}{n}\sum_{i=1}^n \nabla \ell_i(\boldsymbol{\theta}_t)$$

by definition. Then we apply triangle inequality again to obtain

$$\|\nabla L(\boldsymbol{\theta}_t)\| - \frac{1}{n}\sum_{i=1}^n \|\nabla \ell_i(\boldsymbol{\theta}_{t+1}^{(i)})\| \le \left\|\frac{1}{n}\sum_{i=1}^n \nabla \ell_i(\boldsymbol{\theta}_{t+1}^{(i)}) - \nabla L(\boldsymbol{\theta}_t)\right\|. \tag{19}$$

Invoking Assumption 3, stationary condition (14), and combining them with (17), (18), and (19) gives us

$$\|\nabla L(\boldsymbol{\theta}_t)\| \le \frac{\mu}{n}\sum_{i=1}^n \|\boldsymbol{\theta}_{t+1}^{(i)} - \boldsymbol{\theta}_t\| + \sqrt{\lambda_t}c \tag{20}$$

The right-hand-side of (20) can be upperbounded by the norm inequality $\| \cdot \|_1 \leq \sqrt{n} \| \cdot \|_2$ as

$$\sum_{i=1}^{n} \| \boldsymbol{\theta}_{t+1}^{(i)} - \boldsymbol{\theta}_t \| + \frac{\sqrt{\lambda_t} cn}{\mu}$$

$$\leq \sqrt{(n+1) \left( \frac{\lambda_t c^2 n^2}{\mu^2} + \sum_{i=1}^{n} \| \boldsymbol{\theta}_{t+1}^{(i)} - \boldsymbol{\theta}_t \|^2 \right)}$$

$$= \sqrt{n(n+1) \, \mathrm{E}[\| \boldsymbol{\theta}_{t+1} - \boldsymbol{\theta}_t \|^2] + \frac{\lambda_t c^2 n^2 (n+1)}{\mu^2}} \qquad (21)$$

Combining (20) and (21) and squaring both sides shows (16).

$\square$

Simply putting Lemma 1 and Lemma 2 together, we obtain (8) in Proposition 1.

