# OpenReview forum: "Stochastic Proximal Point Algorithm for Large-scale Nonconvex Optimization: Convergence, Implementation, and Application to Neural Networks"
_ICLR.cc/2021/Conference — Reject_

### Official Review · AnonReviewer1 · 2020-10-24
**Analysis seems not rigorous**

**Rating:** 4
**Confidence:** 4

**Review:**

In this paper the authors study stochastic proximal point algorithm for nonconvex optimization, where the model is iteratively updated by solving a proximal optimization problem based on a randomly selected loss function. The authors develop efficient implementation for solving the proximal optimization problem: first for nonlinear least squares and then for general losses. Then the authors study the convergence rates for the developed algorithm. Upper bounds on the expected average squared gradients are developed for both constant step sizes and diminishing step sizes. Experimental results are also reported to support the algorithm in practical implementations.

Comments.

1. The authors show that the proximal optimization problem can be efficiently solved. This is nice. However, the idea in the development of the algorithm seems standard. It seems a bit surprising that this algorithm has not been developed before.

2. The convergence rates are a bit surprising. For example if we set $\lambda=0$ in Thm 2, then eq (9) shows that the averaged gradient converges to zero, which should not happen since in this case the algorithm makes no progress.

3. In Appendix A, the authors make two assumptions in (13) and (14). However, it remains unclear whether these two assumptions can be satisfied simultaneously. In particular, does the stationary point in (14) satisfies the sufficient decrease in (13)?

4. I think eq (20) is not correct. The term $\sqrt{\lambda_t}c$ should be $c/\lambda_t$. As the theoretical results depend on this inequality, the results are not correct.

5. In Assumption 3, the authors assume the updates lie in a compact set. This can be only guaranteed if you impose a constraint on the space. However, the constraint would make the stationarity in (14) no longer hold.

6. In Theorem 1, the equation is not complete.

7. In eq (10), $\alpha$ should be $\alpha_t$

---

### Official Review · AnonReviewer4 · 2020-10-28
**A badly-written paper**

**Rating:** 3
**Confidence:** 4

**Review:**

This paper considers the stochastic proximal point algorithm for solving nonconvex nonlinear least squares optimization problems. A linearization strategy is used to accelerate the procedure and in each iteration the algorithm works by solving a linear system. Some convergence analysis for the proposed is present. Some experiments have been conducted.

I have the following comments.
1. In the proposed algorithm, the authors only take one example instead of a batch of training examples to construct the gradient. This strategy often results in much large variance and slow convergence in practice.

2. The results in Proposition 1 does not imply the convergence of the algorithm. The theoretical analysis is incremental.

3. The numerical comparisons are not sufficient.  The authors should include the comparisons with state-of-the-art second-order optimization solver such as K-FAC.

---

### Official Review · AnonReviewer2 · 2020-10-28
**Insufficient theoretical and experimental results**

**Rating:** 3
**Confidence:** 4

**Review:**

This paper studies the stochastic proximal point algorithm (SPPA) for large-scale nonconvex optimization problems. The authors propose to use Gauss-Newton to perform the proximal update in nonlinear least squares and L-BFGS or accelerated gradient for generic problems. The authors derive the convergence of SPPA to a stationary point in expectation for nonconvex problems, and perform numerical experiments to showcase the effectiveness of the proposed method compared to SGD and its variants.

The paper is generally clear, yet the convergence analysis is mainly based on adapting Bottou et al. (2018). While the proposed methods could be significant additions to stochastic optimizers in deep learning, I found the study of the current paper is insufficient; see the cons below.

Pros:
- It is well-known that the proximal point algorithm (PPA) converges faster than gradient descent (GD), and the same holds for their stochastic counterparts. One advantage is that the step sizes in PPA and SPPA can be larger than those in GD and SGD, which can speed up convergence. The proximal steps in PPA and SPPA are however hard to perform for generic (nonconvex) problems since the proximity operator of the objective functions usually do not have closed forms. The proposal of the authors to perform efficient inner-loop optimization schemes like Gauss-Newton and L-BFGS allows approximation of such proximal steps, without much computational burden added.

Cons:
- Theory:
    * I found that Assumption 3 is too strong and do not think it is a standard assumption. Otherwise the constant $ c $ can be very large. This also leads to a question of why using the upper bound $ c $ is the second part of the RHS of (20) but not the first part?
    * Also why $ \sqrt{\lambda_t} $ instead of $ \lambda_t $ in (20)? If I did not misunderstand, it is derived from (14).
    * As $ c $ and hence $ C $ can be very large, the bounds (9) and (10) in Theorems 2 and 3 can well be vacuous.
    * Also the quantifier in Assumption is missing (for all $ i\in \lbrace 1, \ldots, n \rbrace $?)
    * Another pitfall of the theoretical results of this work is that the convergence analyses of the proposed SPPA-LBFGS, SPPA-AGD and SPPA-GN are all missing, especially since this work considers nonconvex problems.

- Experiments:
    * To showcase the proposed methods are really comparable to or outperform methods like SGD or Adam, numerical experiments should be performed on data sets of larger scales and much deeper networks. In particular, the regression data sets in the paper are so small that the gain of the proposed method over other baselines are so marginal.

Typos:
- Theorem 1: do you mean the limit is equal to 0 or is finite? I guess something is missing.
- Proof of Theorem 2: $ L(\theta_1) - \mathbb{E}[L(\theta_{T+1})] $ instead of $ L(\theta_0) - \mathbb{E}[L(\theta_{T})] $
- (10): the LHS of the inequality should be $ \alpha_t $ instead of $ \alpha $

---

### Official Review · AnonReviewer3 · 2020-11-04
**the novelty is not enough**

**Rating:** 4
**Confidence:** 5

**Review:**

This paper revisits the stochastic proximal point algorithm (SPPA) and apply SPPA to solve nonconvex optimization problems with efficient subproblem solvers.

Firstly, there is a work [Chen et. al 2020] which also provides the convergence result of SPPA on manifold problem and it is not the weakly convex setting.

Secondly, the convergence rate of SPPA is 1/epsilon^2, which is the same as SGD. Regarding the convergence result, there is no advantage of SPPA against SGD and the author should have a discussion about this. The convergence rate is asymptotic and this paper does not point out which iteration we should use as the final output.

Thirdly, the convergence analysis is rather standard and lack of novelty. Moreover, the convergence result of Gauss-Newton and L-BFGS to solve the proximal subproblem should also be provided. Since the main concern for PPA-type method lies on the convergence behavior and efficiency to solve the proximal subproblem. Furthermore, in the experiment part, the comparison of running time between SPPA and SGD, ADAM, Adagrad should also be provided.

Lastly, I wonder whether the assumption 1 is reasonable, since I have not seen this assumption in other nonconvex stochastic programming papers. Authors should remark on this assumption and it would be better to put some references on this.


Confidence level: 5, abusolutely certain.

Rating: reject

references:
[1] Manifold Proximal Point Algorithms for Dual Principal Component Pursuit and Orthogonal Dictionary Learning.
Shixiang Chen, Zengde Deng, Shiqian Ma, Anthony Man-Cho So.  arXiv preprint arXiv:2005.02356, 2020.

---

### Decision · Program_Chairs · 2021-01-07
**Final Decision**

**Decision:**

Reject

**Comment:**

All reviewers recommend rejection: concerns were raised in terms of technical correctness, quality of presentation and the quality of experiments. There was no rebuttal. The AC agrees with the reviewers and recommends rejection.